# The impact of antenatal syphilis point of care testing on pregnancy outcomes: A systematic review

**Dana Brandenburger[1], Elena Ambrosino[2]***

1 Faculty of Health, Medicine & Life Sciences, University of Maastricht, Maastricht, The Netherlands,
2 Department of Genetics and Cell Biology, Research School GROW (School for Oncology & Development), Institute for Public Health Genomics (IPHG), Faculty of Health, Medicine & Life Sciences, University of Maastricht, Maastricht, The Netherlands

* e.ambrosino@maastrichtuniversity.nl

**Data Availability Statement:** All relevant data are within the manuscript and its Supporting Information files.

## Abstract

### Background

Mother-to-child transmission of syphilis remains a leading cause of neonatal death and still-birth, disproportionally affecting women in low-resource settings where syphilis prevalence rates are high and testing rates low. Recently developed syphilis point-of-care tests (POCTs) are promising alternatives to conventional laboratory screening in low-resource settings as they do not require a laboratory setting, intensive technical training and yield results in 10–15 minutes thereby enabling both diagnosis and treatment in a single visit. Aim of this review was to provide clarity on the benefits of different POCTs and assess whether the implementation of syphilis POCTs is associated with decreased numbers of syphilis-related adverse pregnancy outcomes.

### Methods

Following the PRISMA guidelines, three electronic databases (PubMed, Medline (Ovid), Cochrane) were systematically searched for intervention studies and cost-effectiveness analyses investigating the association between antenatal syphilis POCT and pregnancy outcomes such as congenital syphilis, low birth weight, prematurity, miscarriage, stillbirth as well as perinatal, fetal or infant death.

### Results

Nine out of 278 initially identified articles were included, consisting of two clinical studies and seven modelling studies. Studies compared the effect on pregnancy outcomes of treponemal POCT, non-treponemal POCT and dual POCT to laboratory screening and no screening program. Based on the clinical studies, significantly higher testing and treatment rates, as well as a significant reduction (93%) in adverse pregnancy outcomes was reported for treponemal POCT compared to laboratory screening. Compared to no screening and laboratory screening, modelling studies assumed higher treatment rates for POCT and predicted the most prevented adverse pregnancy outcomes for treponemal POCT, followed by a dual treponemal and non-treponemal POCT strategy.

**Funding:** The authors received no specific funding for this work.

**Competing interests:** Authors have no conflict of interest, financial or otherwise.

## Conclusion

Implementation of treponemal POCT in low-resource settings increases syphilis testing and treatment rates and prevents the most syphilis-related adverse pregnancy outcomes compared to no screening, laboratory screening, non-treponemal POCT and dual POCT. Regarding the benefits of dual POCT, more research is needed. Overall, this review provides evidence on the contribution of treponemal POCT to healthier pregnancies and contributes greater clarity on the impact of diverse diagnostic methods available for the detection of syphilis.

## 1. Introduction

Syphilis is a sexually-transmitted disease (STD) caused by the spirochete bacterium *Treponema pallidum* which can be effectively treated with a single long-acting dose of penicillin [1–4]. With approximately 6 million global new cases annually, it is one of the most prevalent STDs [5]. During pregnancy, the infection can be vertically transmitted to the fetus resulting in severe adverse pregnancy outcomes such as congenital syphilis, fetal loss or stillbirth, neonatal death and prematurity or low birth weight [6]. In 2016, more than half a million cases of congenital syphilis were recorded globally, which resulted in more than 200.000 stillbirths and neonatal deaths. As long as effective diagnosis and treatment are provided in an early stage of pregnancy, congenital syphilis and other adverse syphilis-related pregnancy outcomes are efficiently preventable and treatable [7]. Still, congenital syphilis remains the second most common cause of preventable stillbirth, only outnumbered by malaria, and disproportionally affects women in low-resource settings [5, 8].

Syphilis develops in different stages: early syphilis (primary, secondary and early latent syphilis) and late syphilis (latent and tertiary syphilis). Vertical transmission to the fetus through the placenta is possible in every stage of infection and gestation [9]. However, transmission depends on the extent of spirochetes present in the blood and therefore the risk of mother-to-child transmission (MTCT) is highest is early syphilis, especially during the secondary stage [9]. Because manifestations of different syphilis stages are often (primary syphilis) or always (latent syphilis) asymptomatic, if not diagnosed using screening methods the infection frequently remains unrecognized [4], which increases the MTCT risk [10].

In 2007, the WHO commenced a global initiative for the elimination of congenital syphilis [11]. Even though syphilis screening of pregnant women at their first antenatal care (ANC) contact is recommended in almost all countries globally, transmission of syphilis from mother-to-child remains a public health problem and pregnant women often undiagnosed and untreated, despite low costs of efficient diagnosis and medications [7, 12, 13]. Furthermore, syphilis disproportionally affects women in low-resource settings where prevalence rates are high and testing rates are extremely low, as is the case of Nigeria and the Democratic Republic of Congo, where more than 3% of women are infected with syphilis but only 2% and 16% are screened, respectively [14–16]. Therefore, it is vital to scale up syphilis screening programs for pregnant women, particularly those suitable for low-resource settings.

The current gold-standard method for the diagnosis of syphilis is a combination of serologic laboratory based non-treponemal test and treponemal tests [4, 17]. Non-treponemal tests detect antibodies produced in response to lipoidal material released during syphilis-related cell damage and revert to negative test results after successful treatment [18]. However, their

applicability is limited in early primary and late syphilis due to their low sensitivity, frequently resulting in false-negative results. Additionally, false-positive non-treponemal test results have been reported with ongoing co-infections, such as tuberculosis, malaria or hepatitis C infection [19]. Therefore, non-treponemal tests are usually combined with treponemal tests. Treponemal tests detect antibodies to *T. pallidum* proteins which remain detectable after successful treatment and thus remain positive for life making the distinction between current and previous infections difficult, often leading to overtreatment of women with past infections [20]. For effective diagnosis both tests are performed sequentially either with the traditional algorithm (positive non-treponemal tests result confirmed with a treponemal test) or with the reverse algorithm (positive treponemal test, followed by non-treponemal test) [19]. An overview of the described laboratory screening algorithms can be found in S1 Fig.

Because laboratory tests require technical expertise, equipment, electricity and refrigeration, they are often inaccessible in resource-limited settings, which poses a serious problem for diagnosis and treatment in a population where the burden of maternal syphilis remains a serious challenge [8, 21, 22]. A recently developed promising alternative are rapid point-of-care tests (POCT), which constitute a clear advantage in resource-limited settings as they yield results in 10–15 minutes, do not require a laboratory setting or intensive technical training and can be stored at room temperature [4, 21, 23]. Furthermore, the risk of patients lost to follow up, which is particularly high in resource-limited areas, is reduced as patients can receive both diagnosis and treatment in a single visit [4]. At present, a variety of POCTs are available, of which several fulfil the ASSURED criteria, developed by the WHO to assess the Affordability, Sensitivity, Specificity, User-friendliness, Rapidity and robustness, Equipment-freeness and Delivery to the end user of tests, features which are crucial in low-resource settings [24]. Most of the POCTs that meet the ASSURED criteria are immunochromatographic strips (ICS) treponemal tests, but new POCTs have been launched recently that are a combination of treponemal and non-treponemal tests [3].

Several studies demonstrated a significant increase in the proportion of syphilis screening and same day treatment for ANC attendees to >90% in low- and middle-income countries (LMICs) such as Brazil, Peru, Tanzania, Uganda, Zambia and China after the introduction of POCTs [25]. Yet, challenges remain in the implementation of POCTs, particularly in resource-limited settings, such as the acceptance of local healthcare workers, provision of effective treatment, regular supply of test kits and quality assurance [4]. Furthermore, as POCTs for syphilis, particularly non-treponemal tests and treponemal combined tests, so-called non-treponemal and treponemal dual POCTs, are relatively new [26], only limited data on the effect of their use on pregnancy outcomes is currently available. The aim of this study was to investigate whether the implementation of different types of antenatal POCTs positively correlates with decreased numbers of syphilis-related adverse pregnancy outcomes and contributes to healthier pregnancies. In addition, since diagnostic methods to detect syphilis are heterogenous and since the health impact of novel POCTs still lacks evidence, this study seeks to provide greater clarity on the benefits of the variety of different syphilis tests to pregnant women and their children in low-resource settings.

## 2. Methods

A systematic literature search was performed following the Preferred Reporting Items for Systematic Review and Meta-Analysis Statement (PRISMA) guidelines [27, 28]. The corresponding completed PRISMA form can be found in S1 Checklist. Due to the studies' heterogeneity, particularly in terms of testing options and outcomes measured, a systematic literature review, but no meta-analysis, was conducted.

## 2.1 Eligibility criteria

Studies investigating the impact of antenatal syphilis POCTs on pregnancy outcomes were eligible for this review. POCTs were defined as medical diagnostic tools that yield results rapidly (20–30 minutes) allowing diagnosis and treatment in one single visit. Studies were only included if participants were diagnosed using syphilis POCTs during pregnancy and pregnancy outcomes were identified either through diagnosis or a prediction model. Since previous research suggests that adverse pregnancy outcomes associated with maternal syphilis infection can be reliably predicted from published data on disease prevalence, ANC coverage, treatment rates and screening and testing effectiveness, and as clinical studies on the effect of POCT on syphilis-related pregnancy outcomes are relatively scarce, prediction models were considered a valid source of information [7, 29]. Intervention studies as well as cost-effectiveness analyses were eligible for this review. (Systematic) reviews, case reports and surveys were excluded. Furthermore, only human-based and English written studies with no restriction on publication date were included.

## 2.2 Outcome measurements

Adverse pregnancy outcomes related to syphilis infection included: congenital syphilis, low birth weight, prematurity, miscarriage and stillbirth, as well as perinatal, fetal or infant death. Reported adverse pregnancy outcomes were only considered relevant if related to the ongoing pregnancy.

## 2.3 Search strategy

The literature search was conducted in three electronic databases, PubMed, Medline (Ovid) and Cochrane and included all literature published as of June 8th, 2020. Medical Subject Headings (MeSH) terms and free text terms combined with Boolean (AND, OR) terms were used. The search strategy was developed and tested in PubMed and consequently adapted for the other two databases and can be found in S1 Table. Relevant keywords were: "syphilis", "treponema pallidum", "syphilis infected women", "point-of-care testing", "point-of-care systems", "point-of-care diagnostics", "rapid testing", "pregnancy", "pregnant", "antenatal", "prenatal", "pregnant women", "pregnancy outcome", "congenital syphilis", "stillbirth", "perinatal death", "low birth weight", "fetal death", "prematurity", "mortality", "death", "spontaneous abortion", "pregnancy complications", "neonatal death", "infant death", "perinatal mortality", "clinical evidence of syphilis". Duplicates were excluded using the bibliographic management software EndNote.

## 2.4 Study selection

Articles were assessed by means of their title and abstract by one researcher (DB). Afterwards, the full text of potential articles was read and assessed for their eligibility according to the inclusion/exclusion criteria. In cases of ambiguity, a second researcher (EA) was consulted. Additionally, bibliographies of potentially relevant papers, even if they were excluded during the selection process, were screened for additional potential studies.

## 2.5 Data extraction

To investigate the association between syphilis POCTs and pregnancy outcomes, primary variables of interests were the tests used to detect syphilis and resulting pregnancy outcomes. Retrieved articles were grouped into clinical and modelling studies. Relevant data was extracted on authors, country, study design and population, number and proportion of

pregnant women tested, details on syphilis tests used, syphilis prevalence in ANC setting, treatment delay and type of treatment, as well as number and type of pregnancy outcomes. Additionally, maternal and gestational age were extracted from clinical studies and critical model input parameters including risk of specific pregnancy outcomes for healthy, treated and untreated mothers were obtained from modelling studies. Prevalence of specific pregnancy outcomes was calculated per 1.000 pregnancies for each individual study.

## 2.6 Assessment of methodological quality of selected studies

The Joanna Briggs Institute (JBI) Critical Appraisal Tool was used to evaluate the methodological quality of selected studies [30]. For each individual study, a respective JBI checklist, developed for different study designs and containing questions to assess the potential risk of bias, was applied. Questions that were answered with "yes" were assigned 1 point, "no" 0 points and "unclear" 0.5 points. After completion of the checklist, points were summed up for every individual study. The risk of bias was considered to be "low" for studies that reached 70% or higher of the maximum number of points (11 points for cost-effectiveness analyses, 10 for randomized controlled trials), "moderate" for studies with a result between 50% and 69% and "high" for studies scoring 49% or lower [30–32]. The critical appraisal checklist as provided by the Joanna Briggs Institute Reviewer´s Manual can be found in S2 Table. The risk of bias was not used to support the exclusion of studies from this review, following what is customary in systematic reviews.

## 3. Results

### 3.1 Study selection

From an initial 278 articles identified, nine studies were eligible for inclusion, including seven modelling studies and two clinical studies. The PRISMA flow diagram describing the steps of study selection can be found in Fig 1.

### 3.2 Risk of bias within studies

All clinical studies and five modelling studies had a low risk of bias (study score above 70%), while two modelling studies had a moderate risk of bias (study score between 50% and 69%) (S3 Table). Both randomized controlled trials had a risk of selection bias since outcome assessors were not blinded to treatment groups, and it remained unclear if true randomization had occurred in the study conducted by Munkhuu et al. [33, 34]. Furthermore, all cost-effectiveness analyses lacked adjustments of costs and outcomes for differential timing, and a comprehensive description of alternative tests was lacking for four studies [23, 35–37]. Five cost-effectiveness studies had missing costs, either regarding pregnancy outcome costs [35, 36, 38] or patient costs such as travel, waiting and treatment time [35, 36, 38, 39]. Three modelling studies were missing treatment rates [23, 35, 36] and one study lacked a well-defined question as described in the JBI critical appraisal tool (S2 and S3 Tables) [40].

### 3.3 Clinical studies

**3.3.1 Population and study characteristics.** In total, two clinical studies were retrieved, one from Mongolia, published in 2009, and one from South Africa, published in 2003 [33, 34]. Both studies were cluster-randomized trials with cohorts of pregnant women attending their first ANC visit [33, 34]. Study characteristics, including details on cohorts and methodological features, can be found in Table 1. The aim of both studies was to compare the effect of the implementation of syphilis POCTs in ANC settings to conventional laboratory testing. While

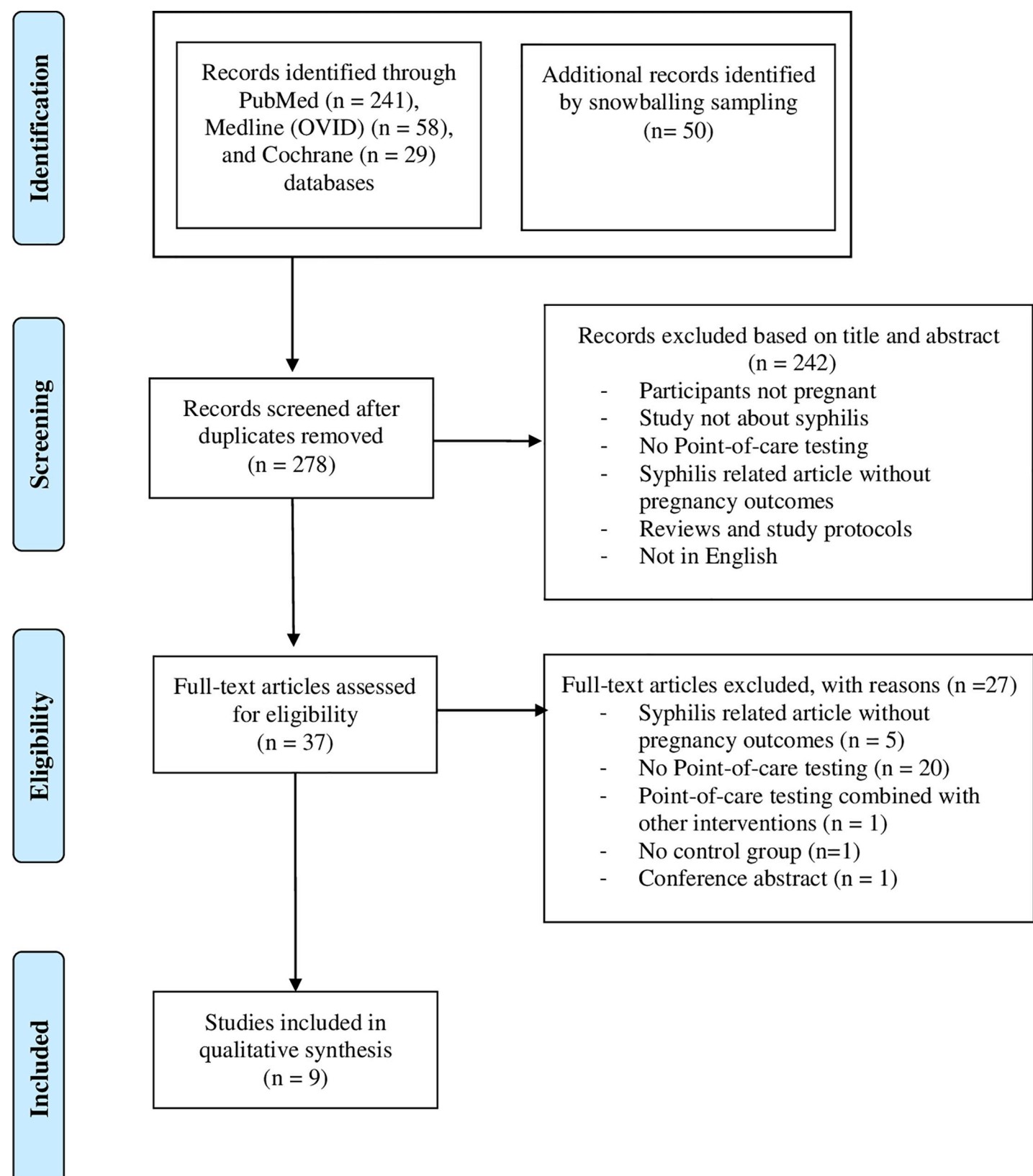

**Fig 1. PRISMA based flow diagram displaying the study selection.**

**Table 1. Study characteristics in clinical studies.**

| Author, year | *Munkhuu et al. 2009* [34] | *Myer et al. 2003* [33] |
|---|---|---|
| Country | Mongolia | South Africa |
| Study design | cluster randomized controlled trial | cluster randomized controlled trial |
| Study population | pregnant women attending at first ANC visit | pregnant women attending at first ANC visit |
| Number of participants | 7700 | 7134 |
| Follow-up | followed up to point of delivery | followed up to point of delivery |
| Intervention group—method used to detect syphilis | POC syphilis testing (SD Bioline Syphilis 3.0) at first visit, at third GA semester and after delivery | onsite RPR test |
| Confirmatory test for positive POCT patients | confirmatory RPR+TPHA laboratory test of positive POCT patients | confirmatory RPR laboratory test of positive POCT patients |
| Control group–method used to detect syphilis | RPR+TPHA laboratory screening | RPR laboratory screening |
| Sensitivity | NR | onsite RPR: 62% increased to 83% for women with titres greater than 1:4 |
| | | laboratory RPR: NR |
| Specificity | NR | onsite RPR: 96% |
| | | laboratory RPR: NR |
| Mean maternal age, years (SD) | POCT: 26,9 (5,5) | onsite RPR: 25,8 |
| | RPR+TPHA: 27 (7,5) | laboratory RPR: 27,0 |
| GA at sampling at first ANC visit, weeks (SD) | at first sampling | onsite RPR: 23,7 |
| | POCT: 14,1 (6,6) | laboratory RPR: 24,2 |
| | RPR+TPHA: 12 (4,8) | |
| Women receiving antenatal syphilis screening | POCT: 1st test: 99,9% | NR |
| | 2nd test: 99.7% | |
| | RPR+TPHA: 1st test: 79.9% | |
| | 2nd test: 62.1% | |
| | (significant difference between POCT and control group, p <0.001) | |
| Syphilis prevalence in antenatal care setting | POCT: 1st test: 1.9%, 2nd test: 0.5% | 7.5% for both onsite and laboratory RPR |
| | RPR+TPHA: 1st test: 0.9%, 2nd test: 0.08% | |
| Treatment | 3 doses 2.4 MU benzathine penicillin injection | 3 doses 2.4 benzathine penicillin injection |
| % receiving adequate treatment | POCT: 98.9% | onsite RPR: 64.1% |
| | RPR+TPHA: 89.6% | laboratory RPR: 68% |
| | (p = 0.02) | (difference not significant) |
| Treatment delay | POCT: same day treatment | onsite RPR: same day treatment |
| | laboratory RPR+TPHA test: treatment at first follow-up visit, time period not reported | laboratory RPR: treatment at first follow-up visit |
| | | mean difference in treatment delay after onsite and laboratory RPR test: 16,4 days (significant) |
| Type of pregnancy outcomes | CS | MC, PND |

ANC, Antenatal care. CI, confidence interval. CS, congenital syphilis. FL, fetal loss. GA, gestational age. ICS, Immunochromatographic strip. LBW, low birth weight. LMIC, low- and middle-income country. MC, miscarriage. NND, neonatal death. NR, not reported. PM, prematurity. PND, perinatal death. RPR, rapid plasma reagin. SB, stillbirth. ST, standard deviation. TPHA, Treponema pallidum particle agglutination assay.

Munkhuu et al. [34] used a treponemal POCT (SD Bioline Syphilis 3.0), Myer and colleagues [33] implemented an onsite rapid plasma reagin (RPR) test, which is sensitive to nontreponemal antibodies and was performed on bedside with battery powered equipment. Both studies confirmed positive POCT results with a laboratory syphilis test. Authors compared their results to a control group which underwent conventional laboratory screening. The proportion of women receiving syphilis screening and the number of syphilis cases detected was

**Table 2. Association between syphilis POCTs and pregnancy outcomes in clinical studies.**

| Author, year | Intervention group—POCT used to detect syphilis | Type of POCT | Control group–method used to detect syphilis | Pregnancy outcome presented in study | Number and type of Pregnancy outcomes per 1.000 pregnancies |
|---|---|---|---|---|---|
| *Munkhuu et al. 2009* [34] | SD Bioline Syphilis 3.0 and confirmatory laboratory RPR/TPHA test of positive POCT patients | treponemal test | laboratory RPR +TPHA | Syphilis screening of 7.700 pregnant women resulted in: | laboratory RPR+TPHA: 1,95 CS cases |
| | | | | RPR/TPHA: 15 CS | SD Bioline Syphilis 3.0: |
| | | | | POCT: 1 CS (reduction of 93%) | 0,13 CS cases (93% reduction p<0.002) |
| *Myer et al. 2003* [33] | onsite RPR and confirmatory RPR laboratory test of positive POCT patients | non-treponemal test | laboratory RPR | Syphilis screening of 723 (561 onsite RPR and 163 off-site RPR) pregnant women resulted in: | laboratory RPR: |
| | | | | laboratory RPR: | 16,6 MC |
| | | | | 12 (3.1%) MC | 24,9 PND |
| | | | | 18 (5.1%) PND | onsite RPR: |
| | | | | onsite RPR: | 6,91 MC (reduction of 58%) |
| | | | | 5 (2.1%) MC | 11,06 PND (reduction of 55%) |
| | | | | 8 (3.3%) PND | (Difference not significant, P = 0.31)) |

CS, congenital syphilis. MC, miscarriage. PND, perinatal death. POCT, point-of-care testing. RPR, rapid plasma reagin. TPHA, Treponema pallidum particle agglutination assay.

significantly higher for women in the POCT group, compared to laboratory testing in the study by Munkhuu et al. [34] (99% versus 62.1% - 79.9% and 1.9% versus 0.9%). In contrast, Myer at al. [33] did not document the proportion of women receiving screening and reported the same syphilis prevalence between groups (7.5%). Treatment in both studies consisted of three 2.4 million units (MU) benzathine penicillin injections. First dose of treatment was administered on the same day as POCT in intervention groups, and at the first follow up visit for control groups. Mean treatment delay was only reported by Myer et al. [33] (Table 1). The proportion of women receiving adequate treatment was significantly higher for POCT patients in the study by Munkhuu et al. [34], while it was the same between groups for the study by Myer et al. [33]. Gestational age (GA) at screening at first ANC visit varied between 14 and 24 weeks (second trimester) for Myer et al. [33] and Munkhuu et al. [34] respectively in the intervention group, whereas laboratory syphilis testing in the latter's control group occurred at 12 weeks GA (first trimester).

**3.3.2 Association between syphilis POCTs and pregnancy outcomes.** Of the two clinical studies included, Munkhuu et al. [34] reported congenital syphilis as main pregnancy outcome, whereas Myer and colleagues [33] reported miscarriage and perinatal death. Congenital syphilis was defined if any of the following existed: classic sign of congenital syphilis in neonate, mother with syphilis lesion at delivery, untreated mother with positive syphilis test at delivery, treponemas seen in autopsy material or neonate with RPR titers at least four-fold than maternal titers [34]. Perinatal death was defined either as stillbirth (child born dead at or after 28 weeks GA) or early neonatal death (death up to 7 days postpartum). Miscarriage was not further defined [33]. Table 2 provides a summary of the association between syphilis testing and pregnancy outcomes in clinical studies. Munkhuu et al. [34] documented a significant 93% reduction of congenital syphilis cases in the treponemal POCT group compared to conventional laboratory testing (1,95 and 0,13 congenital syphilis cases per 1.000 pregnancies in the laboratory screening group and POCT group, respectively), indicating a positive association between treponemal POCT and healthy pregnancy outcomes. By implementing a non-

treponemal rapid RPR, the study by Myer et al. [33] described a 58% reduction of miscarriage and 55% reduction of perinatal deaths (16,6 versus 1,91 miscarriages and 24,9 versus 11,06 perinatal deaths per 1.000 pregnancies in the laboratory screening group and POCT group, respectively), however this difference was not statistically significant.

### 3.4 Modelling studies

**3.4.1 Population and study characteristics.** In total, seven modelling studies, published between 2007 and 2016, were included in the analysis. All studies were cost-effectiveness analyses predicting the effect of different syphilis POCTs on pregnancy outcomes for a study population of pregnant women with access to at least one ANC visit. Details on study characteristics and methodological features can be found in Table 3. Five studies had cohorts based in sub-Saharan Africa, of which two included the entire region [35, 40], two were based in South Africa [38, 39] and one in Malawi [23]. The remaining two studies were based in Latin America and Asia, of which one study focused on Haiti [37] and one on 11 Asian and 20 LMICs [36]. Syphilis prevalence in ANC settings varied between 0.1% and 14%, depending on study and country. Four studies predicted outcomes for treponemal POCTs alone [23, 35–37], two studies focused on both a treponemal immunochromatographic strip test (ICS) and a non-treponemal POCT (onsite RPR) [38, 39] and one study predicted pregnancy outcomes for a treponemal (ICS), a non-treponemal (onsite RPR) and a treponemal and non-treponemal dual POCT [40]. All studies compared the effect of POCTs to a control group consisting either of no screening [35, 36], conventional laboratory RPR+TPHA screening [23], both no screening and laboratory RPR+TPHA screening [38–40] or both syndromic surveillance and laboratory RPR screening [37]. Sensitivity and specificity of the different POCTs and the comparator laboratory tests assumed in the prediction model were retrieved from published literature, differed per study and can be found in Table 3. Sensitivity and specificity of treponemal, non-treponemal and laboratory RPR+TPHA screening did not vary considerably between studies. Only Blandford et al. [38] made a distinction regarding test specificity for women with past syphilis infections, which reduced the specificity to 11% for treponemal POCTs, and early and late maternal syphilis which reduced the sensitivity of non-treponemal onsite RPR to 39% for late maternal syphilis. Furthermore, Rydzak et al. [39] made a distinction between syphilis stages which reduced the sensitivity of laboratory RPR+TPHA testing to 66% and 69% for primary syphilis and late latent syphilis, respectively. To model expected pregnancy outcomes, authors implemented the risk of specific pregnancy outcomes for untreated mothers and/or treated mothers and/or mothers without syphilis as model input parameters. The risk of adverse pregnancy outcomes for untreated mothers was calculated by all studies [23, 35–40]. Additionally, five studies calculated the risk for mothers without syphilis [23, 35, 37, 39, 40] and three studies the risk for treated mothers [35–37]. Studies that only calculated one of those two, assumed that a treated mother had the same risk as a mother without syphilis (Table 4, Section risk of pregnancy outcome). Treatment was assumed to consist of either three injections [35–37] or one injection of 2.4 MU benzathine penicillin [23, 38–40]. All studies assumed same day treatment for POCT patients and a treatment delay of 1–2 weeks for laboratory screened women. Five modelling studies reported treatment rates for both POCTs and laboratory screening, which varied between 87% and 100% and between 58.8% and 67% for POCTs and laboratory RPR+TPHA screening, respectively [37–40]. One study reported treatment rates of 80% for laboratory RPR+TPHA testing which was relatively high compared to the other studies, and did not report treatment rates for women receiving POCTs [23]. The retrieved modelling studies reported five types of adverse pregnancy outcomes namely congenital syphilis, stillbirth, neonatal death, low birth weight and miscarriage (Table 3). All

**Table 3. Study characteristics in modelling studies.**

| Author, year | *Kuznik et al. 2013* [35] | *Kuznik et al. 2015* [36] | *Schackman et al. 2007* [37] | *Blandford et al. 2007* [38] | *Rydzak et al. 2008* [39] | *Owusu-Edusei et al. 2011* [40] | *Bristow et al. 2016* [23] |
|---|---|---|---|---|---|---|---|
| **Country** | 43 countries in sub-Saharan Africa | 11 Asian and 20 Latin American Countries (LMICs) | Haiti | Rural Eastern Cape Province, South Africa | South-Africa | sub-Saharan Africa | Malawi |
| **Study design** | cost effectiveness analysis | cost effectiveness analysis | cost effectiveness analysis | cost effectiveness analysis | cost effectiveness analysis | cost effectiveness analysis | cost effectiveness analysis |
| **Study population** | pregnant women with access to at least one ANC visit | pregnant women with access to at least one ANC visit | pregnant women with access to at least one ANC visit | pregnant women with access to at least one ANC visit | pregnant women with access to at least one ANC visit | pregnant women with access to at least one ANC visit | pregnant women with access to at least one ANC visit |
| **Number of participants** | 23,5 million | 47,2 million women in Asia and 10,1 million in Latin America | 202.000 (168.000 in rural areas and 35.000 in urban areas) | 1.000 | 1000 women with 6 pregnancies over lifetime resulting in 6.000 pregnancies | 1.000 | 100.000 |
| **Intervention group—POCT used to detect syphilis** | ICS | ICS | Determine Syphilis TP<br><br>SD Bioline Syphilis 3.0<br><br>Omega VisiTect Syphilis | onsite RPR<br><br>ICS | onsite RPR<br><br>ICS | onsite dual POCT<br><br>ICS<br><br>onsite RPR | Omega VisiTect Syphilis |
| **Control group–method used to detect syphilis** | no screening | no screening | syndromic surveillance<br><br>laboratory RPR | no screening<br><br>laboratory RPR +TPHA | no screening<br><br>laboratory RPR +TPHA | no screening<br><br>laboratory RPR +TPHA | laboratory RPR +TPHA |
| **Sensitivity (CI, when provided)** | ICS: 86% (74.5–94.1%) | ICS: 86% (74.5–94.1%) | rapid tests: 83.3%<br><br>laboratory RPR: 75.6% | ICS: 100% for early maternal syphilis, 86% for late maternal syphilis<br><br>onsite RPR: 71% for early maternal syphilis, 39% for late maternal syphilis<br><br>onsite RPR: 71% for early maternal syphilis, 39% for late maternal syphilis | onsite RPR:<br><br>primary syphilis 77%<br><br>secondary syphilis 99%<br><br>early latent syphilis 99%<br><br>late latent syphilis 70%<br><br>ICS:<br><br>primary syphilis 82.0%<br><br>all other syphilis stages 98.3%<br><br>laboratory RPR +TPHA: primary syphilis: 65.9%<br><br>secondary & early latent syphilis: 98%<br><br>late latent syphilis: 69.3% | dual POCT: 88.6%<br><br>ICS: 98%<br><br>onsite RPR: 71%<br><br>laboratory RPR +TPHA: 100% | POCT: 82%<br><br>laboratory RPR +TPHA: 100% |

*(Continued)*

**Table 3.** (Continued)

| Author, year | Kuznik et al. 2013 [35] | Kuznik et al. 2015 [36] | Schackman et al. 2007 [37] | Blandford et al. 2007 [38] | Rydzak et al. 2008 [39] | Owusu-Edusei et al. 2011 [40] | Bristow et al. 2016 [23] |
|---|---|---|---|---|---|---|---|
| **Specificity (CI, when provided)** | ICS: 99% (97.8–99.7%) | ICS: 99% (97.8–99.7%) | rapid tests: 98.9% | ICS: 99% for never-infected women, 11% for women with past disease | onsite RPR: 96.4% | dual POCT: 98% | POCT: 96% |
| | | | laboratory RPR: 95.7% | onsite RPR: 98% for never-infected women, 95% for women with past disease | ICS: 94.1% | ICS: 94% | laboratory RPR +TPHA: 100% |
| | | | | laboratory RPR +TPHA: 100% for never-infected women and women with past disease | laboratory RPR +TPHA: 100% | onsite RPR: 98%<br>laboratory RPR +TPHA: 100% | |
| **Risk of pregnancy outcome–mother without syphilis** | SB: 4.6%<br>NND: NR | NR | SB: 9.3%<br>NND: 2.2% | NR | SB: 1%<br>NND: 1%<br>LBW: 9%<br>MC 1st trimester: 12%<br>MC 2nd trimester: 1%<br>Healthy: 89% | SB: 1%<br>NND: 1%<br>LBW: 9%<br>MC: 1%<br>Healthy: 89% | SB/FL: 4.6%<br>NND: 3%<br>PM/ LBW: 6.3% |
| **Risk of pregnancy outcome–mother with syphilis, untreated** | SB: 25.6%<br>NNDs: 12,3<br>CS: 15.5% | SB: 25.6%<br>NND: 12,3<br>CS: 15.5% | SB: 28.2%<br>NND: 15.6%<br>CS: 30.5% | CS: Early maternal infection 94% and late maternal infection: 37% | SB: 11%<br>NND: 4%<br>CS: 60%<br>LBW: 25%<br>MC 1st trimester: 18%<br>MC 2nd trimester: 1.5% | SB: 11%<br>NND: 4%<br>CS: 60%<br>LBW: 25%<br>MC: 1.5% | SB/FL: 25.6%<br>NND: 12.3%<br>CS: 15.5%<br>PM/LBW12.1% |
| **Risk of pregnancy outcome–mother with syphilis, treated** | SB: 10.8%<br>NND: 5.7%<br>CS: 5.2% | SB: 4.6%<br>NND: 2.5%<br>CS: 0.5% | SB: 10.1%<br>NND: 1.1%<br>CS: 1.1% | NR | NR | NR | NR |
| **Syphilis prevalence in antenatal care setting** | 0.6% - 14.0%<br>30 countries < 3.8%<br>10 countries 4% - 8.6%<br>3 countries 10% - 14% | 0.1% - 3.9%<br>Asian countries <1.2%<br>17 Latin American Countries <2%<br>3 Latin American countries: 2.1% - 3.9% | 3.8% in rural setting<br>3,5 in urban setting | 6.3% (of which: 26.6% early and 73.4% late maternal syphilis) | 6% (2% primary or secondary and 4% latent syphilis) | 10% | 1.1%– 2.2% |
| **Women receiving antenatal syphilis screening** | 0% - 93%<br>weighted average: 40.7% | 0.1% - 100%<br>Weighted average: 68.6% | 68% of pregnant women<br>(100% in urban areas and 64% in rural areas) | NR | 79% | NR | 8% |

(*Continued*)

**Table 3.** (Continued)

| Author, year | *Kuznik et al. 2013* [35] | *Kuznik et al. 2015* [36] | *Schackman et al. 2007* [37] | *Blandford et al. 2007* [38] | *Rydzak et al. 2008* [39] | *Owusu-Edusei et al. 2011* [40] | *Bristow et al. 2016* [23] |
|---|---|---|---|---|---|---|---|
| **Treatment** | intervention group: 3 doses 2.4 MU benzathine penicillin injection | intervention group: 3 doses 2.4 MU benzathine penicillin injection | 3 doses 2.4 MU benzathine penicillin injection | 1 dose 2.4 MU benzathine penicillin for early maternal syphilis | 1 dose 2.4 MU benzathine penicillin injection | 1 dose 2.4 MU benzathine penicillin injection | 1 dose 2.4 MU benzathine penicillin injection |
| | control group: no treatment | control group: no treatment | | 3 doses 2.4 MU benzathine penicillin for late maternal syphilis | | | |
| **Treatment delay** | same day treatment | same day treatment | POCT: same day treatment | ICS & onsite RPR: same day treatment | ICS & onsite RPR: same day treatment | ICS, dual POC & onsite RPR: same day treatment | POCT: same day treatment |
| | | | syndromic surveillance: same day treatment | laboratory RPR +TPHA test: treatment at first follow-up visit, time period not reported | laboratory RPR/ TPHA test: treatment at first follow-up visit after | laboratory RPR +TPHA test: treatment at first follow-up visit, time period not reported | laboratory RPR +TPHA test: treatment at first follow-up visit, time period not reported |
| | | | RPR laboratory test: treatment at first follow-up visit, after 1 week | | 2 weeks | | |
| **Treatment rates** | NR | NR | POCT: 100% | POCT: | POCT:87% | POCT: 100% | POCT: NR |
| | | | laboratory RPR +TPHA: | initial treatment: 89% | laboratory RPR +TPHA: 67% | laboratory RPR +TPHA: 67% | laboratory RPR +TPHA: 80% |
| | | | Initial treatment: 58.8% | laboratory RPR +TPHA: initial treatment: 61% | | | |
| **Type of pregnancy outcomes** | SB, NND, CS | SB, NND, CS | SB, NND, CS | CS | CS, LBW, NND, SB | CS, LBW, NND, SB, MC | CS, SB or FL, NND, PM or LBW |

ANC, Antenatal care. CI, confidence interval. CS, congenital syphilis. FL, fetal loss. ICS, Immunochromatographic strip. LBW, low birth weight. LMIC, low- and middle-income country. MC, miscarriage. NND, neonatal death. NR, not reported. PM, prematurity. PND, perinatal death. RPR, rapid plasma reagin. SB, stillbirth. ST, standard deviation. TPHA, Treponema pallidum particle agglutination assay.

studies reported the association between POCTs and congenital syphilis [23, 35–40], six studies looked at neonatal death, stillbirth and POCT [23, 35–37, 39, 40], three studies reported the association between low birth weight and POCT [23, 39, 40], and one study documented the relation between miscarriage and POCT [40]. Since pregnancy outcomes were predicted with a cost-effectiveness analysis, a sensitivity analysis rather than a statistical analysis was performed which displayed the results to be stable over a large range of probabilities.

**3.4.2 Association of treponemal syphilis POCTs and pregnancy outcomes.** A summary of the association of different types of syphilis POCT and pregnancy outcomes can be found in Table 4. Six studies predicted an increased proportion of adverse pregnancy outcomes averted by treponemal POCT, compared to all other screening methods, i.e. no screening, conventional laboratory screening, non-treponemal POCT and dual POCT [35–40]. In contrast, one study predicted no difference in adverse pregnancy outcomes for treponemal POCT screened women compared to laboratory RPR+TPHA screening [23]. The greatest effect of treponemal POCT was observed compared to no screening. Compared to no screening, Kuznik et al. [35, 36] reported 0,82 adverse pregnancy outcomes (of which 0,42 stillbirths, 0,17 neonatal deaths and 0,23 congenital syphilis cases) averted in Latin America, 0,83 (of which 0,43 stillbirths,

**Table 4. Association between syphilis POCTs and pregnancy outcomes in modelling studies.**

| Author, year | Intervention group—POCT used to detect syphilis | Control group—method used to detect syphilis | Pregnancy outcome presented in study | Number and type of Pregnancy outcomes averted per 1.000 pregnancies (annually) |
|---|---|---|---|---|
| | | | *treponemal POCT* | |
| *Kuznik et al. 2013* [35] | ICS | no screening | syphilis screening of 23,5 million pregnant women would result in: | 2,7 SB<br>1,06 NND |
| | | | 64000 SB, 25000 NND, 32000 CS cases averted compared to no screening | 1,36 CS |
| | | | | in total 5,12 adverse pregnancy outcomes averted compared to no screening |
| *Kuznik et al. 2015* [36] | ICS | no screening | syphilis screening of 47,2 million women in Asia and 10,1 million in Latin America would result in: | Asia:<br>0,43 SB<br>0,17 NND |
| | | | Asia: 20344 SB, 8201 NND, 10952 CS cases averted compared to no screening | 0,23 CS |
| | | | | in total 0,83 adverse pregnancy outcomes averted compared to no screening |
| | | | Latin America: 4270 SB, 1721 NND, 2298 CS cases averted compared to no screening | Latin America:<br>0,42 SB<br>0,17 NND<br>0,23 CS |
| | | | | in total 0,82 adverse pregnancy outcomes averted compared to no screening |
| *Blandford et al. 2007* [38] | ICS | no screening | syphilis screening of 1.000 pregnant women would result in: | 27 CS cases averted compared to no screening |
| | | laboratory RPR+TPHA | no screening; 33 CS cases | 11 CS cases averted compared to onsite RPR screening |
| | | onsite RPR | onsite RPR: | 9 CS cases averted compared to laboratory RPR+TPHA screening |
| | | | 17 CS cases (16/33 averted) averts 48% of CS cases that would be expected with no screening program | |
| | | | laboratory RPR+TPHA: | |
| | | | 15 CS cases (18/33 averted) averts 55% of CS cases that would be expected with no screening program | |
| | | | onsite ICS: | |
| | | | 6 CS cases (27/33 averted) averts 82% of CS cases that would be expected with no screening program | |
| *Rydzak et al. 2008°* [39] | ICS | no screening | syphilis screening of 6.000 pregnant women would result in: | 29,68 CS cases<br>7,26 LBW<br>1,41 NND<br>4,87 SB |
| | | laboratory RPR+TPHA | no screening: | in total 43,22 adverse pregnancy outcomes averted compared to no screening |
| | | onsite RPR | 256 CS cases, 583,3 LBW, 70,1 NND, 99,8 SB | |
| | | | laboratory RPR+TPHA: | 9,83 CS cases<br>2,38 LBS<br>0,47 NND<br>1,62 SB |
| | | | 119,1 CS cases, 29,3 LBW, 5,7 NND, 19,5 SB averted compared to no screening and would result in 4915,1 healthy births | in total 14,3 adverse pregnancy outcomes averted compared to laboratory RPR+TPHA screening |
| | | | onsite RPR: | 2,08 CS cases<br>0,48 LBW<br>0,1 NND<br>0,31 SB |
| | | | 165,6 CS cases, 40,7 LBW, 7,9 NND, 27,3 SB compared to no screening and would result in 4993,8 healthy births | |
| | | | onsite ICS: | |
| | | | 178,1 CS cases, 43,6 LBW, 8,5 NND, 29,2 SB averted compared to no screening and would result in 5016,6 healthy births | in total 2,97 adverse pregnancy outcomes averted compared to onsite RPR screening |

(*Continued*)

**Table 4.** (Continued)

| Author, year | Intervention group—POCT used to detect syphilis | Control group–method used to detect syphilis | Pregnancy outcome presented in study | Number and type of Pregnancy outcomes averted per 1,000 pregnancies (annually) |
|---|---|---|---|---|
| *Owusu-Edusei et al. 2011^X* [40] | ICS | no screening | syphilis screening of 1,000 pregnant women would result in: | 17 CS cases |
| | | laboratory RPR+TPHA | | 13 LBW |
| | | | no screening; | 2 NND |
| | | onsite RPR | 39 adverse outcomes: 18 CS cases, 14 LBW, 2 NND, 4 SB, 1 MC | 4 SB |
| | | onsite dual POCT | laboratory RPR+TPHA: | 1 MC |
| | | | 13 adverse outcomes: 6 CS cases, 5 LBW, 1 NND, 1 SB, 0 MC (26 adverse outcomes prevented compared to no screening) | in total 37 adverse pregnancy outcomes averted compared to no screening |
| | | | | 5 CS cases |
| | | | onsite RPR: | 4 LBW |
| | | | 11 adverse outcomes: 5 CS cases, 4 LBW, 1 NND, 1 SB, 0 MC (29 adverse outcomes prevented compared to no screening) | 1 NND |
| | | | | 1 SB |
| | | | | 0 MC |
| | | | onsite dual-POC: | in total 11 adverse pregnancy outcomes averted compared to laboratory RPR+TPHA screening |
| | | | 5 adverse outcomes: 2 CS cases, 2 LBW, 0 NND, 1 SB, 0 MC (34 adverse outcomes prevented compared to no screening) | 4 CS cases |
| | | | | 3 LBW |
| | | | onsite ICS: | 1 NND |
| | | | 2 adverse outcomes: 1 CS cases, 1 LBW, 0 NNDs, 0 SB, 0 MC (37 adverse outcomes prevented compared to no screening) | 1 SB |
| | | | | 0 MC |
| | | | | in total 9 adverse pregnancy outcomes averted compared to onsite-RPR screening |
| | | | | 1 CS cases |
| | | | | 1 LBW |
| | | | | 0 NNDs |
| | | | | 1 SB |
| | | | | 0 MC |
| | | | | in total 3 adverse pregnancy outcomes averted compared to onsite dual POCT screening |
| *Bristow et al. 2016* [23] | Omega VisiTect Syphilis | laboratory RPR+TPHA | syphilis screening of 8,000 pregnant women would result in: | no pregnancy outcomes prevented and 0.001 additional NND compared to laboratory RPR+TPHA screening |
| | | | laboratory RPR+TPHA: | |
| | | | 40 adverse outcomes: 12 CS cases, 16 SB or FL, 7 NNDs, 5 PM/LBW | |
| | | | Omega VisiTect Syphilis: | |
| | | | 41 adverse pregnancy outcomes: | |
| | | | 12 CS cases, 16 SB or FL, 8 NNDs, 5 PM/LBW | |

(*Continued*)

**Table 4.** (Continued)

| Author, year | Intervention group—POCT used to detect syphilis | Control group—method used to detect syphilis | Pregnancy outcome presented in study | Number and type of Pregnancy outcomes averted per 1.000 pregnancies (annually) |
|---|---|---|---|---|
| Schackman et al. 2007 [37] | Determine Syphilis TP | syndromic surveillance | syphilis screening of 202.000 (168.000 in rural areas and 35.000 in urban areas) pregnant women would result in 1.129 CS cases, 786 SB and 437 NNDs averted annually | nationwide: |
| | | | | 5,59 CS cases |
| | SD Bioline Syphilis 3.0 | RPR laboratory test | | 3,90 SB |
| | Omega VisiTect Syphilis | | | 2,16 NNDs |
| | | | in rural setting; compared to syndromic surveillance for every 1.000 women who receive POCT 6 CS cases 6,5 SB and 4,2 NNDs would be averted | in total 11,65 adverse pregnancy outcomes averted compared to syndromic surveillance in rural and RPR laboratory screening in urban settings |
| | | | | rural settings: |
| | | | in urban setting; compared to RPR in lab for every 1.000 women who receive POCT 1,4 CS cases, 1 SB and 1,1 NNDS would be averted | 6 CS cases |
| | | | | 6,5 SB |
| | | | | 4,2 NND |
| | | | | n total 16,7 adverse pregnancy outcomes averted compared to syndromic surveillance |
| | | | | urban settings: |
| | | | | 1,4 CS cases |
| | | | | 1 SB |
| | | | | 1,1 NND |
| | | | | in total 3,5 adverse pregnancy outcomes averted compared to RPR laboratory screening |
| | | | *non-treponemal POCT* | |
| Blandford et al. 2007 [38] | onsite RPR | no screening | syphilis screening of 1.000 pregnant women would result in: | 16 CS cases averted compared to no screening |
| | | laboratory RPR+TPHA | no testing program: 33 CS cases | 2 additional CS cases compared to laboratory RPR+TPHA screening |
| | | ICS | laboratory RPR+TPHA: 15 CS cases (18/33 averted) averts 55% of CS that would be expected with no screening program | 11 additional CS cases compared to ICS screening |
| | | | onsite RPR: 17 CS cases (16/33 averted) averts 48% of CS that would be expected with no screening program | |
| | | | onsite ICS: 6 CS cases (27/33 averted) (averts 82% of CS that would be expected with no screening program) | |
| Rydzak et al. 2008° [39] | onsite RPR | no screening | syphilis screening of 6.000 pregnant women would result in: | 27,6 CS cases |
| | | | | 6,78 LBW |
| | | laboratory RPR+TPHA | no screening: | 1,32 NNDs |
| | | ICS | 256 CS cases, 583,3 LBW, 70,1 NNDs, 99,8 SB | 4,55 SB |
| | | | laboratory RPR+TPHA: | in total 40,25 adverse pregnancy outcomes averted compared to no screening |
| | | | 119,1 CS cases, 29,3 LBW, 5,7 NNDs, 19,5 SB averted compared to no screening | 7,75 CS cases |
| | | | onsite RPR: | 1,9 LBW |
| | | | 165,6 CS cases, 40,7 LBW, 7,9 NNDs, 27,3 SB averted compared to no screening | 0,37 NNDs |
| | | | | 1,3 SB |
| | | | ICS: | in total 11,32 adverse pregnancy outcomes averted compared to laboratory RPR+TPHA screening |
| | | | 178,1 CS cases, 43,6 LBW, 8,5 NNDs, 29,2 SB averted compared to no screening | 2,08 additional CS cases |
| | | | | 0,48 additional LBW |
| | | | | 0,1 additional NNDs |
| | | | | 0,31 additional SB |
| | | | | in total 2,97 additional adverse pregnancy outcomes compared to ICS screening |

*(Continued)*

...

...

**Table 4.** (Continued)

| Author, year | Intervention group—POCT used to detect syphilis | Control group—method used to detect syphilis | Pregnancy outcome presented in study | Number and type of Pregnancy outcomes averted per 1,000 pregnancies (annually) |
|---|---|---|---|---|
| *Owusu-Edusei et al.* 2011[X] [40] | onsite RPR | no screening | syphilis screening of 1,000 pregnant women would result in: | 13 CS cases |
| | | | | 10 LBW |
| | | laboratory RPR+TPHA | no screening: | 1 NND |
| | | | 39 adverse outcomes: 18 CS cases, 14 LBW, 2 NNDs, 4 SB, 1 MC | 3 SB |
| | | dual POCT | | 1 MC |
| | | ICS | | in total 28 adverse pregnancy outcomes averted compared to no screening |
| | | | laboratory RPR+TPHA: | |
| | | | 13 adverse outcomes: 6 CS, 5 LBW, 1 NND, 1 SB, 0 MC (26 adverse outcomes prevented compared to no screening) | 1 CS cases |
| | | | onsite RPR: | 1 LBW |
| | | | 11 adverse outcomes: 5 CS cases, 4 LBW, 1 NND, 1 SB, 0 MC (29 prevented compared to no screening) | 0 NNDs |
| | | | | 0 SB |
| | | | | 0 MC |
| | | | onsite dual-POC: | in total 2 adverse pregnancy outcomes averted compared to laboratory RPR+TPHA screening |
| | | | 5 adverse outcomes (2 CS cases, 2 LBW, 0 NND, 1 SB, 0 MC (34 prevented compared to no screening) | 3 additional CS cases |
| | | | ICS: | |
| | | | 2 adverse outcomes: 1 CS cases, 1 LBW, 0 NNDs, 0 SB, 0 MC (37 prevented compared to no screening) | 2 additional LBW |
| | | | | 1 additional NND |
| | | | | 0 additional SB |
| | | | | 0 additional MC |
| | | | | in total 6 additional adverse pregnancy outcomes compared to dual POCT screening |
| | | | | 4 additional CS cases |
| | | | | 3 additional LBW |
| | | | | 1 additional NND |
| | | | | 1 additional SB |
| | | | | 0 additional MC |
| | | | | in total 9 additional adverse pregnancy outcomes compared to ICS screening |
| | | | *dual-treponemal & non-treponemal syphilis POCT* | |

*(Continued)*

**Table 4.** (Continued)

| Author, year | Intervention group—POCT used to detect syphilis | Control group—method used to detect syphilis | Pregnancy outcome presented in study | Number and type of Pregnancy outcomes averted per 1,000 pregnancies (annually) |
|---|---|---|---|---|
| Owusu-Edusei et al. 2011[X] [40] | Dual Path Platform (DDP®) Syphilis Test (Chemio Diagnostic Systems, Inc) | no screening | syphilis screening of 1,000 pregnant women would result in: | 16 CS cases |
| | | | | 12 LBW |
| | | | | 2 NNDs |
| | | laboratory RPR+TPHA | | 3 SB |
| | | | | 1 MC |
| | | | no screening: | in total 34 adverse pregnancy outcomes averted compared to no screening |
| | | onsite RPR | 39 adverse outcomes: 18 CS cases, 14 LBW, 2 NNDs, 4 SB, 1 MC | 4 CS cases |
| | | | laboratory RPR+TPHA: | 3 LBW |
| | | ICS | 13 adverse outcomes: 6 CS cases, 5 LBW, 1 NND, 1 SB, 0 MC (26 adverse outcomes prevented compared to no screening) | 1 NND |
| | | | | 0 SB |
| | | | onsite RPR: | 1 MC |
| | | | 11 adverse outcomes: 5 CS cases, 4 LBW, 1 NND, 1 SB, 0 MC (29 prevented compared to no screening) | in total 9 adverse pregnancy outcomes averted compared to laboratory RPR+TPHA screening |
| | | | onsite dual-POCT: | RPR+TPHA screening |
| | | | 5 adverse outcomes: 2 CS cases, 2 LBW, 0 NND, 1 SB, 0 MC (34 prevented compared to no screening) | 3 CS cases |
| | | | ICS: | 2 LBW |
| | | | 2 adverse outcomes: 1 CS case, 1 LBW, 0 NNDs, 0 SB, 0 MC (37 prevented compared to no screening) | 1 NND |
| | | | | 0 SB |
| | | | | 0 MC |
| | | | | in total 6 adverse pregnancy outcomes averted compared to onsite RPR screening |
| | | | | 1 additional CS cases |
| | | | | 1 additional LBW |
| | | | | 0 additional NNDs |
| | | | | 1 additional SB |
| | | | | 0 additional MC |
| | | | | in total 3 additional adverse pregnancy outcomes compared to ICS screening |

CS, congenital syphilis. FL, fetal loss. ICS, Immunochromatographic strip. LBW, low birth weight. MC, miscarriage. NND, neonatal death. PM, prematurity. PND, perinatal death. RPR, rapid plasma reagin. SB, stillbirth. TPHA, Treponema pallidum particle agglutination assay.

\*\*X these data pertain to the same study.

0,17 neonatal deaths and 0,23 congenital syphilis cases) in Asia [36] and 5,12 (of which 2,70 stillbirths, 1,06 neonatal deaths, 1,36 congenital syphilis cases) in sub-Saharan Africa for every 1.000 pregnancies [35]. Also based in sub-Saharan Africa, two other studies predicted 27 prevented congenital syphilis cases in South Africa [38] and 37 averted adverse pregnancy outcomes (17 congenital syphilis cases, 13 cases of low birth weight, 2 neonatal deaths, 4 stillbirths, 1 miscarriage) in the entire region of sub-Saharan Africa per 1.000 pregnancies for women receiving treponemal POCT [40]. Likewise based in South Africa, one study reported 43,22 adverse pregnancy outcomes prevented per 1.000 pregnancies, compared to no screening intervention [39]. Schackman et al. [37] differentiated between treponemal POCT implementation in rural and urban areas of Haiti and predicted 16,70 adverse pregnancy outcomes (per 1.000 pregnancies) prevented, compared to syndromic surveillance, the conventional method in rural settings in Haiti, and 3,50 adverse pregnancy outcomes (per 1.000 pregnancies) averted, compared to conventional RPR laboratory screening in urban areas. Of the four other studies which compared treponemal POCT to laboratory RPR+TPHA screening, three authors predicted a larger proportion of adverse pregnancy outcomes prevented for treponemal POCTs [38–40]. This effect, however, was smaller than compared to no screening intervention. Specifically, compared to laboratory RPR+TPHA screening, 11 congenital syphilis cases were averted per 1.000 pregnancies (as opposed to 27 congenital syphilis cases compared to no screening) [38], 14,30 adverse pregnancy outcomes were prevented (in contrast to 43,22 adverse outcomes averted compared to no screening) and 11 adverse pregnancy outcomes were prevented (in contrast to 37 adverse pregnancy outcomes averted compared to no screening) [40]. The only study that did not predict beneficial effects of treponemal POCTs was the one conducted by Bristow et al. [23], which reported the same proportion of congenital syphilis cases, stillbirth or fetal loss and prematurity or low birth weight for both treponemal POCTs and laboratory RPR+TPHA screening. The same study even reported an 0,001 additional neonatal deaths per 1.000 pregnancies for women receiving syphilis POCTs.

**3.4.3 Association of non-treponemal syphilis POCTs and pregnancy outcomes.** Of the three studies investigating the implementation of non-treponemal onsite RPR tests, all predicted an increased proportion of adverse pregnancy outcomes averted compared to no screening, resulting in a decreased prevalence of adverse pregnancy outcomes for non-treponemal onsite RPR screened women [38–40] (Table 4). Compared to treponemal POCT however, onsite RPR prevented fewer adverse pregnancy outcomes in all three retrieved studies and resulted in 11 additional congenital syphilis cases in one study [38] and 2,97 additional adverse pregnancy outcomes in another study, both based in South Africa [39], and 9 additional adverse pregnancy outcomes (all per 1.000 pregnancies) in a study considering the entire sub-Saharan Africa [40]. Two studies predicted that non-treponemal POCTs would prevent more adverse pregnancy outcomes compared to laboratory RPR+TPHA screening, and prevented 11,32 adverse pregnancy outcomes according to Rydzak et al. [39] or 2 adverse pregnancy outcomes according to Owusu-Edusei et al. [40], per 1.000 pregnancies. In contrast, another study reported no beneficial effect compared to laboratory RPR+TPHA screening, and reported 2 additional congenital syphilis cases per 1.000 pregnancies for onsite RPR screened women [38].

**3.4.4 Association of dual-treponemal & non-treponemal syphilis POCTs and pregnancy outcomes.** Only one study predicted pregnancy outcomes for women screened with a dual-treponemal and non-treponemal POCT in a cohort based in sub-Saharan Africa [40]. Compared to no screening, laboratory RPR+TPHA and onsite RPR screening, dual POCT prevented 34, 9 and 6 pregnancy outcomes respectively, for every 1.000 pregnancies. However, no beneficial effect was reported compared to treponemal ICS screening which would prevent 3 additional adverse pregnancy outcomes to dual POCT [40].

## 4. Discussion

This study aimed to assess the evidence on the association between different types of antenatal syphilis POCTs and syphilis-related pregnancy outcomes and provide greater clarity on the impact of diverse diagnostic methods available for the detection of syphilis in pregnancy.

### 4.1 The impact of treponemal POCTs on adverse pregnancy outcomes

Of the one clinical [34] and seven modelling studies that reported the association between treponemal POCTs and syphilis-related pregnancy outcomes, all [35–40] except one [23] reported that this approach averted the most adverse pregnancy outcomes compared to no screening, laboratory RPR+TPHA screening and non-treponemal POCTs. These findings suggest that the implementation of treponemal POCTs in ANC settings in which pregnant women receive no screening, laboratory screening or non-treponemal POCTs prevents adverse pregnancy outcomes such as stillbirth, neonatal deaths, congenital syphilis, low birth weight and miscarriage, and results in healthier pregnancies.

The only included study that did not demonstrate a positive impact of implementing treponemal syphilis POCTs on pregnancy outcomes was the study conducted by Bristow et al. [23]. For a cohort of pregnant women receiving syphilis screening in Malawi, Bristow et al. [23] predicted the same proportion of adverse pregnancy outcomes for treponemal POCTs and laboratory screening. Since the proportion of averted pregnancy outcomes depends on the assumed treatment rates implemented, this might be attributed to the assumed treatment rates the authors used in the prediction model. Indeed, in contrast to the other modelling studies included, Bristow et al. [23] did not report the treatment rate for women receiving POCTs, and implemented a relatively high treatment rate for women receiving laboratory screening, compared to other included modelling studies and previously published clinical studies from the same region (80% versus 61% - 67% in other studies) [38–41]. If the treatment rates for POCTs and laboratory screening were to be similar, this in turn could have influenced the impact of syphilis POCTs on pregnancy outcomes.

In contrast, in the included clinical study by Munkhuu et al. [34], the authors documented along with a decreased prevalence of adverse pregnancy outcomes, significantly higher testing and treatment rates for women who received treponemal POCTs. This is in line with the assumed treatment rates of the included modelling studies and with a previous clinical study conducted in Peru, which documented a significant increase from 82% to 99% and 39% to 95% for testing and treatment after implementation of POCTs in a setting where laboratory RPR+TPHA screening was the conventional testing method [42]. Compared to the other modelling studies, Kuznik et al. [35, 36] observed only relatively small, yet beneficial effects of treponemal POCTs compared to no screening intervention in both their studies. This might be attributed to several reasons. Firstly, each modelling study included in this review used a unique prediction model which consequently impacts resulting pregnancy outcomes. Secondly, since the proportion of averted pregnancy outcomes depends on the assumed treatment rates implemented in the prediction model which were not reported in either of the two studies by Kuznik et al. [35, 36], it remains uncertain what proportion of women tested was assumed to receive treatment. Thirdly, authors assumed relatively low syphilis prevalence rates (30/43 countries from sub-Saharan Africa with prevalence below 3.8%, 10/43 countries between 4% and 8.6% and 3/43 countries ≥ 10%), compared to other included studies that were based in the same region [35]. For example, while other included studies assumed a syphilis prevalence of 6% in South Africa [38, 39] and 10% in sub-Saharan Africa in general [40], Kuznik et al. [35] assumed a syphilis prevalence of 1.9% in South Africa and only reported a syphilis prevalence of ≥ 10% in 3/43 countries in sub-Saharan Africa. Consequently, a model

which assumes relatively low syphilis prevalence, would predict fewer (averted) adverse pregnancy outcomes. In line with this, included modelling studies that assumed an ANC setting with relatively high syphilis prevalence, also predicted a greater effect on syphilis-related pregnancy outcomes [38–40]. One shortcoming in both studies by Kuznik et al. [35, 36] was the control group authors implemented. While other studies also included a laboratory screening strategy [23, 37–40], Kuznik et al. [35, 36] only compared the effect of treponemal POCTs to no screening intervention. However, since laboratory syphilis screening is usually the conventional method in settings where a syphilis screening strategy is in place [17, 43], it would have been valuable to compare the impact of syphilis POCT on pregnancy outcomes to laboratory screening.

## 4.2 The impact of non-treponemal POCTs on adverse pregnancy outcomes

In the present study, three modelling and one clinical study reported the implementation of onsite-RPR (non-treponemal POCT) and syphilis-related pregnancy outcomes [33, 38–40]. All three modelling studies reported healthier pregnancies after the implementation of onsite-RPR in settings where no syphilis screening programme is in place. However, this effect was smaller, with fewer adverse pregnancy outcomes averted compared to women receiving treponemal POCT [38–40]. Compared to settings with laboratory screening programmes, two studies predicted better pregnancy outcomes for women receiving onsite-RPR [39, 40], while one reported healthier pregnancy outcomes for the conventional laboratory screening method [38]. One reason why Blandford et al. [38] might have predicted more favourable pregnancy results for laboratory screening could be the sensitivity and specificity the authors used in their model. Blandford et al. [38] implemented 100% sensitivity and specificity for laboratory screening and were the only included authors who differentiated between early and late maternal syphilis which decreased the sensitivity of the onsite RPR for late maternal syphilis to 39% in their model. This disparity in sensitivity between laboratory screening and onsite RPR might have resulted in more favourable pregnancy results for women receiving laboratory RPR+TPHA. The sensitivity of the onsite RPR used by Blandford et al. [38] is in line with previously conducted clinical studies that demonstrated relatively low sensitivity for onsite-RPR (71% for high-titer syphilis and 39% for low-titer syphilis) [41]. This indicates that the two authors who did not differentiate between early and late maternal syphilis and predicted healthier pregnancy outcomes for women receiving onsite-RPR might have overestimated the sensitivity of onsite RPR and its impact on pregnancy outcomes [39, 40]. The second clinical study included compared the impact of onsite-RPR on pregnancy outcomes with the effects of laboratory RPR+TPHA screening [33]. Even though Myer et al. [33] documented an approximately 50% decrease of syphilis-related pregnancy outcomes compared to conventional laboratory screening, this reduction was not significant. Similarly, Myer et al. [33] did not document a change in treatment rates, despite reducing treatment delay for POCT patients considerably. There are several possible reasons why the implementation of onsite-RPR did not demonstrate a significant impact on pregnancy outcomes. On the one hand, the frequency of adverse pregnancy outcomes was lower than expected in the control group compared to baseline, resulting in the same proportion of women receiving treatment. On the other hand, authors documented no increase in treatment rates for women receiving POCT due to technical and logistical difficulties, as the onsite-RPR is relatively complex to perform [33]. These findings resemble previous experiences from primary care settings in LMICs, where nurses reported onsite-RPR to be time consuming, as well as relatively difficult to read and perform since serum needs to be separated from blood cells with a rotator and mixed with antigens manually [33, 41]. Besides that, problems in upkeeping a regular supply of reagents and

batteries for rotators in primary care settings have been documented frequently [41]. Given that non-treponemal tests also suffer from relatively weak sensitivity, especially during early primary and late syphilis which increases the risk of false-negative results [44], these experiences suggest that onsite-RPR would prevent syphilis-related adverse pregnancy outcomes in settings without other syphilis screening interventions. However, treponemal POCTs would be the preferable screening method in all settings and the benefit of onsite-RPR in situations where a well-functioning laboratory screening programme is already established remains ambiguous.

### 4.3 The impact of dual-treponemal and non-treponemal POCTs on adverse pregnancy outcomes

Only one included modelling study determined the impact of a dual-treponemal & non-treponemal syphilis POCT on syphilis-related pregnancy outcomes [40]. In settings where no screening, laboratory screening or onsite RPR screening programs are offered to pregnant women, Owusu et al. [40] demonstrated a favourable impact of dual POCTs on pregnancy outcomes, predicting that dual POCTs would prevent the most adverse pregnancy outcomes compared to the other three screening algorithms. However, this positive effect was smaller than for treponemal ICS POCTs, with lower numbers of total adverse pregnancy outcomes prevented. Against the authors expectations, the implementation of treponemal POCTs resulted in a greater proportion of healthy pregnancies than the dual POCT strategy [40]. One possible reason could be the considerably higher sensitivity of treponemal POCTs assumed in the author´s prediction model (98% for treponemal-only POCTs versus 88% for dual POCTs). A more recently published meta-analysis and a clinical study conducted in China demonstrating the benefit of combined treponemal and non-treponemal syphilis POCTs also documented a slight decrease of sensitivity for the non-treponemal component of dual POCTs, since it is less sensitive to low-titer RPRs [45, 46]. Nevertheless, authors of both studies still documented good sensitivity (90.1% - 98.2% for the treponemal component and 80.6% - 98.2% for the non-treponemal component) and specificity (91% - 98% for the treponemal and 89.4% for the non-treponemal component) for the dual POCT strategy [45, 46]. By combining the high sensitivity of treponemal POCTs and the ability of non-treponemal tests to differentiate between previous and past syphilis infections, authors of both studies suggest that the dual syphilis POCT strategy has the potential to accurately detect current syphilis infections and reduce overtreatment rates of women with previously treated syphilis infection [45]. Contrary to these findings, another field study implementing a dual syphilis POCT strategy in Burkina Faso reported no reduction of overtreatment for women receiving dual POCTs and an increased proportion of women that remained undiagnosed, compared to women receiving only treponemal POCT. This was due to decreased sensitivity of dual POCTs [47]. Since all of these previously conducted studies reported a decreased sensitivity for dual POCTs compared to treponemal POCTs, and Langendorf et al. [47] documented no improvement in overtreatment rates, the benefits of dual POCTs compared to treponemal POCTs remain unclear. Therefore, further research is needed to confirm the added value of dual POCTs for the rapid diagnosis of syphilis.

### 4.4 Comparison of the effect of treponemal, non-treponemal and dual syphilis POCTs

Overall, it becomes apparent that each of the included studies is flawed in its own way and predicted pregnancy outcomes highly depend on assumed syphilis prevalence, treatment rates and sensitivity and specificity of tests. Nevertheless, the present results demonstrate that the

implementation of treponemal POCTs would be advantageous in LMICs preventing syphilis-related pregnancy outcomes and resulting in healthier pregnancies, independent of the current screening methods in place. Findings regarding the benefits of onsite RPR (non-treponemal POCTs) were mixed and demonstrated only modest improvements of testing and treatment rate and a greater risk of false-negative testing results due to decreased sensitivity, especially during early primary and late syphilis. Furthermore, findings of both clinical studies suggest that resulting pregnancy outcomes are highly dependent on increased testing and treatment rates. While one included clinical study documented a critical impact of implementing treponemal syphilis POCTs on testing and treatment rates [34], the other included clinical study only recorded a reduction in treatment delay for women receiving onsite RPR [33]. These results agree with previous studies conducted in LMICs that demonstrated significantly higher testing and treatment rates for treponemal POCTs (ICS test), compared to both laboratory RPR+TPHA and onsite RPR screening in South Africa, Tanzania, Uganda, Zambia Haiti, Brazil, Peru and China [25, 41, 42, 48]. Previous studies conducted in several LMICs reported favourable acceptability and feasibility ratings of treponemal POCTs among ANC staff [25, 49]. In contrast to onsite RPR, health care workers described the treponemal POCT as easy to perform, and documented greater daily testing capacity with POCT, than with conventional laboratory screening programmes [25, 33, 41]. Among ANC attendees, POCT acceptability was very high and 99.9% of women documented a preference of receiving testing and treatment in a single visit, agreed to wait at the hospital for their test results and favoured finger pricks over venepuncture [25, 49]. Yet the introduction of POCTs in resource-limited settings has been reported challenging for understaffed facilities and often overworked ANC staff. Clinics in resource-limited settings often face challenges such as supply shortages of testing material or providing adequate training to health care workers. Outlining the importance of good training, a study conducted in Mozambique demonstrated greater accuracy of POCTs when conducted by laboratory staff with intensive training, rather than when conducted by health care workers on bedside. Further, Balira et al. [50] documented that only 25% of ANC staff in Tanzania received training in the prevention of syphilis mother-to-child transmission, in general [51]. Additionally, as countries begin to introduce syphilis POCTs, adequate quality assurance programmes must be established, which have been the norm for most laboratory syphilis screening programs but have frequently been neglected for POCTs [4]. Despite these challenges, the findings of this review suggest that the implementation of treponemal syphilis POCTs or dual treponemal and non-treponemal syphilis POCTs in LMIC ANC settings, where no syphilis screening program, laboratory screening or onsite RPR screening are in place, could increase both testing and treatment rates, consequently resulting in fewer syphilis-related pregnancy outcomes and healthier pregnancies. While syphilis screening of pregnant women frequently remains inadequate in low-resource settings, HIV screening programs in ANC settings are already more advanced due to their higher international priority and financial support resulting in 40–50 percentage points higher HIV testing rates, compared to syphilis screening rates in countries such as India, Uganda and Ethiopia [16, 52]. Given that syphilis and HIV coinfections are common and a recently published systematic review of the global prevalence of sexually transmitted co-infections estimated that more than 9% of HIV-positive adults are coinfected with syphilis, ANC programs for simultaneous screening interventions of HIV and syphilis are a promising opportunity to improve syphilis screening [52, 53]. Several studies investigating the effect of syphilis POCT implementation into established HIV screening programs, documented significantly increased syphilis screening rates, like for example from 4% to 95% in Kenya, high preference of patients for dual HIV and syphilis POCTs and excellent laboratory and field performance of dual POCTs for detection of treponemal and HIV antibodies [52, 54–56]. Therefore, implementation of treponemal or dual treponemal and

nontreponemal POCTs into already established HIV screening programs, might be a promising way to improve clinical practice in low-resource settings, reduce syphilis-related adverse pregnancy outcomes and contribute to the WHO Global Health Sector Strategy´s efforts to reduce syphilis incidence globally [57].

### 4.5 Limitations

The present review is subject to certain limitations. Of the nine studies included in this review only two were clinical studies, whereas seven were modelling studies. The results of modelling studies are only as good as the input parameters used to construct the model. Since different modelling studies retrieved their input values from different sources, parameters for test sensitivity and specificity, risk of specific pregnancy outcomes and country specific parameters, like testing and treatment rates as well as treatment delay contributed to the heterogeneity of the included data. Additionally, for some included modelling studies it remained unclear whether seroprevalence consisted of active syphilis infection, as it was uncertain which tests were used to generate local syphilis prevalence data and some prevalence estimates might have been generated using treponemal-only methods, which would not indicate a measure of active syphilis infection, but rather a measure of past or current infection [36]. Therefore, the assumed syphilis prevalence might not have been accurate in all studies.

Furthermore, several modelling studies might have overestimated the sensitivity and specificity of laboratory testing, as well as of POCTs. For example three modelling studies did assume a perfect (100%) sensitivity of RPR+TPHA screening [23, 38, 40], which is contrary to previous research demonstrating lower sensitivity (75.7%) for laboratory RPR+TPHA screening, and might have therefore overestimated the benefit of conventional laboratory testing [58]. Additionally, studies that predicted pregnancy outcomes for POCTs might have overestimated the sensitivity and specificity of tests, as they frequently use manufacturer-provided sensitivities and specificities, which are indicative of a laboratory atmosphere but not of "real word" scenarios in the field. This is for example true for the SD Bioline syphilis POCT, for which a sensitivity and specificity of 83.3% and 98.9% respectively was implemented in the modelling study by Schackman et al. [37]. When implemented by end-users in field conditions in South Africa however, sensitivity and specificity of the SD Bioline POCT were only 66.7% and 98.0%, respectively [59]. Also, a possible bias overestimating the impact of POCTs might have been introduced by studies that assumed 100% treatment rates for women receiving POCTs, as cases of women leaving before receiving treatment have been described previously [33]. Furthermore, challenges such as supply shortages, stockouts, lack of trained health care workers and quality control, which have been reported from field experiences in Africa, were not considered in the modelling studies that would decrease the positive effect of POCTs [60, 61]. Lastly, because dual treponemal and non-treponemal POCTs have been recently developed, only one included study determined the association between dual POCTs and syphilis-related pregnancy outcomes.

## 5. Conclusion

As mother-to-child transmission of syphilis remains a leading cause of neonatal death and stillbirth and disproportionally affects women in low-resource settings where syphilis prevalence rates are particularly high and screening rates low, an overview of the impact of different syphilis POCTs on syphilis-related pregnancy outcomes is crucial to improve maternal and new-born healthcare in low-resource settings [5, 8, 14–16].

Overall, this review demonstrates that the implementation of treponemal POCTs increases testing and treatment rates of pregnant women and is associated with healthier pregnancies in

LMIC settings, where no screening strategy, laboratory screening or non-treponemal POCT programs are in place. Particularly promising, as they detect both treponemal and non-treponemal antibodies, are new dual syphilis POCTs that meet the WHO ASSURED criteria and have been introduced only recently [3, 24]. Unfortunately, through the systematic search of three databases only one study investigating the impact of dual treponemal and non-treponemal POCTs was retrieved and included in this review. Since dual POCTs are still relatively new, research on the feasibility of dual treponemal and nontreponemal POCTs is still relatively limited and benefits compared to treponemal POCTs alone remain ambiguous. Therefore, it would be of interest, to focus future research on the effect of the implementation of dual treponemal and non-treponemal POCTs on syphilis-related pregnancy outcomes, as well as testing and treatment rates [62]. Furthermore, studies implementing syphilis POCTs in established HIV screening programs did show promising results, suggesting a possible way to efficiently introduce syphilis POCT screening in low-resource settings [52, 54–56].

Overall, this review provides greater clarity on the heterogenous diagnostic methods available for the detection of syphilis in pregnancy, and provides evidence on the contribution of treponemal and dual POCTs to healthier pregnancies. By this, this work will pave the way to improved syphilis screening programs and clinical practice in low-resource settings and contribute to the WHO Global Health Sector Strategy´s efforts to reduce syphilis incidence globally [57].

## Supporting information

**S1 Checklist. PRISMA checklist displaying the page numbers where the section or topic is provided [32].**
(DOCX)

**S1 Fig.** (A) Laboratory testing algorithms for the diagnosis of syphilis. RPR, rapid plasma reagin. VDLR, veneral disease research laboratory. TPHA, *treponema pallidum* heamoagglutination assay. TPPA, *treponema pallidum* particle agglutination [62, 63]. (B) Point-of-care testing algorithms for the diagnosis of syphilis. ICS, immunochromatographic strip. POCT, point-of-care testing. RPR, rapid plasma reagin [63].
(DOCX)

**S1 Table. Search strategies and hits.** Based on searches last conducted on June 8, 2020 in PubMed, Medline (Ovid) and Cochrane.
(DOCX)

**S2 Table. Questions used for the critical appraisal for economic evaluations and randomized controlled trials as provided by Joanna Briggs Institute Reviewer´s Manual [30].**
(DOCX)

**S3 Table. Results of the Joanna Briggs Institute critical appraisal checklist.** Q: Questions based on the JBL risk assessment (Appendix 3). ✓: Indicates yes (1 point). O: Indicates No (0 points). '?': Indicates unclear (0,5 points). Risk[b]: The risk of bias was considered high when the study score ≤ 49%, moderate when the study score reached 50 to 69%, and low when the study score reached ≥ 70%. N/A = not applicable.
(DOCX)

## Author Contributions

**Conceptualization:** Elena Ambrosino.

**Data curation:** Dana Brandenburger.

**Formal analysis:** Dana Brandenburger.

**Methodology:** Dana Brandenburger, Elena Ambrosino.

**Supervision:** Elena Ambrosino.

**Writing – original draft:** Dana Brandenburger, Elena Ambrosino.

**Writing – review & editing:** Dana Brandenburger, Elena Ambrosino.

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
