## [Decision Letter · Decision Letter 0]

23 Dec 2020

PONE-D-20-29965

The impact of antenatal syphilis point of care testing on pregnancy outcomes: a systematic review

PLOS ONE

Dear Dr. Ambrosino,

Thank you for submitting your manuscript to PLOS ONE. After careful consideration, we feel that it has merit but does not fully meet PLOS ONE’s publication criteria as it currently stands. Therefore, we invite you to submit a revised version of the manuscript that addresses the points raised during the review process.

We look forward to receiving your revised manuscript.

Kind regards,

Jodie Dionne-Odom, MD

Academic Editor

PLOS ONE

Journal Requirements:

2. Please ensure that you have addressed all items recommended in the PRISMA checklist and include the referenced (but not present) Figure 1: flow chart in the manuscript or as supplemental materials.

4. Please upload a copy of Figure 1, to which you refer in your text on page 9. If the figure is no longer to be included as part of the submission please remove all reference to it within the text.

5. Please include your tables as part of your main manuscript and remove the individual files. Please note that supplementary tables should be uploaded as separate "supporting information" files.

7. We note that this manuscript is a systematic review or meta-analysis; our author guidelines therefore require that you use PRISMA guidance to help improve reporting quality of this type of study. Please upload copies of the completed PRISMA checklist as Supporting Information with a file name “PRISMA checklist”.

Reviewers' comments:

Reviewer's Responses to Questions

**Comments to the Author**

1. Is the manuscript technically sound, and do the data support the conclusions?

Reviewer #1: No

Reviewer #2: Yes

Reviewer #3: Yes

2. Has the statistical analysis been performed appropriately and rigorously? 

Reviewer #1: No

Reviewer #2: Yes

Reviewer #3: N/A

3. Have the authors made all data underlying the findings in their manuscript fully available?

Reviewer #1: Yes

Reviewer #2: Yes

Reviewer #3: Yes

4. Is the manuscript presented in an intelligible fashion and written in standard English?

Reviewer #1: Yes

Reviewer #2: Yes

Reviewer #3: Yes

5. Review Comments to the Author

Reviewer #1: please see comments below- no comments for the authors at this time

I have significant concerns about the study design. The authors selected 2 clinical studies with different outcomes and 7 modeling studies for the analysis. I would suggest a statistical review for this. I don't believe they should use CEA or modeling studies for a systematic review. The two clinical studies had different outcomes that were measured, so I am not sure a meta-synthesis is appropriate. I will be willing to re-review the manuscript if a formal statistical review suggests this is acceptable scientific method.

Reviewer #2: This is an excellent systemic review of studies on the impact of antenatal syphilis point of care testing on pregnancy outcomes. The authors carefully selected the literature, meticulously reviewed the papers, and accurately reported the synthetic results.

Reviewer #3: This manuscript, "The impact of antenatal syphilis point of care testing on pregnancy outcomes: a systematic review" is a very thoughtful and comprehensive analysis of the impact of point of care syphilis tests on adverse pregnancy outcomes such as infant prematurity, stillbirth, and neonatal death. There is not a great deal of literature in this space, but the authors did an excellent job of summarizing what IS available. Their methodology is strong, and the write-up is clear. My overall recommendations are really two-fold: consider shortening the Introduction section a bit (to get to the heart of your paper faster) and consider editing the Discussion section slightly to help the reader understand your main message more quickly (the main message I am seeing is: "each of these studies is flawed in its own way.")

Below are my requests for minor revisions:

1) I found the number of acronyms in the manuscript to be overwhelming and to impede general comprehension. Some acronyms (e.g., POCT) are quite natural given their common use in this literature, but I found other acronyms (e.g., PM for prematurity, FL for fetal loss, NTT for non-treponemal test, ICS for ?) to be neither space-saving nor comprehension-improving. There is no need to abbreviate a one-word term like prematurity. There's also no advantage to introducing separate acronyms for STD and STI when one would likely suffice throughout. Please review all acronyms and reduce the total number to something more reasonable. An acronym is only advantageous when it saves a significant number of words AND when it doesn't require the reader to thumb through the manuscript looking for its original meaning.

2) It would be helpful if the authors introduced the dual POCT test as a 'screen and confirm' or 'non-treponemal and treponemal' dual test (at first use). The phrase 'dual test' is often used to reference HIV/syphilis combination tests, so the clarification might be helpful for a broader audience.

3) The Introduction section feels quite long. I'm already a bit tired by the time I get to the most interesting parts of the manuscript--Methods & Results. I would recommend making some cuts in the Introduction to improve its punchiness and get to the 'so what?' of your work a little faster. For consideration: Cut the sentence beginning with "This initiative was followed by..." on Line 60. We don't need as much history about WHO's strategies to understand your work. Cut the sentences that give examples of traditional lab tests (i.e., TPPA, TPHA). Knowing the names of specific treponemal (bench) tests does not help me understand your point-of-care testing work any better. Cut the entire paragraph about the future of PCR testing. This paragraph is not needed to set up the importance of your POCT work. The sentence beginning on Line 107--which lists WHO's POCT requirements--feel very redundant to everything you already laid out in lines 103-106. Find a way to reduce this redundancy. Consider finding a way to introduce a paragraph break somewhere in lines 60-93 as well (suggested in Line 78) to help with the visual flow of that very long paragraph.

4) Comments about prevalence and case-count in the first paragraph of the Introduction need a qualifier (i.e., 'globally' or 'in the world') so the reader understands what geography you're referencing (Line 42 and 45).

5) Line 51: Early syphilis also includes early latent syphilis (not just primary and secondary). Please add to your parenthetical.

6) Add a citation to Lines 52-53 (vertical transmission comment).

7) Line 64: "fewer," not "less." (There are several instances of this throughout the manuscript that should be checked).

8) Line 119 would read more clearly as: "Yet challenges remain in the implementation of POCT, particularly in resource-limited settings, such as acceptance by local healthcare workers..."

9) Lines 130-134: This paragraph feels like it could be in the Discussion section. This is a strong recommendation (but not a requirement) that you consider relocating this paragraph. It seems to make more sense to tout the contributions (potential or real) of your work AFTER you've shared the results, not before.

10) Line 140: I had never heard/used the term 'meta-synthesis' before. It struck me as a little odd. If this term is preferred by PLOSOne, then I defer. Otherwise, consider using the phrase 'systematic literature review,' which is what this really is and will leave your readers with less head-scratching. If you'd like to keep the phrase 'meta-synthesis' then it might be helpful to provide some basic features of a meta-synthesis that make it different from a systematic literature review.

11) Consider including treatment coverage in your list of 'predictors' in Lines 148-152.

12) Line 159: "...low birth weight, prematurity, miscarriage, and stillbirth..."

13) Instead of saying 'up until June 8th,' it might be more appropriate to say that the search was conducted in 3 databases and included all literature published as of June 8th.

14) Line 179: "were screened" not "where screened."

15) Line 188: "...maternal and gestational age were extracted..."

16) Methods: Add a comment to your methods section (likely to last paragraph of Methods) that addresses whether the bias scores will be used for inclusion/exclusion purposes. It doesn't look like any studies were dropped because of moderate bias scores, but it might be helpful to have you comment as to why that is (or what's customary when using these bias scores).

17) Line 246: Would it be appropriate to replace the phrase 'sampling' with 'first ANC visit?' This would read more intuitively if it's accurate. It reminds the reader that these are real tests whose use is intended to be 'screening at first ANC visit' in most places. If this is not an accurate description of how women were sampled, then you can ignore this recommendation.

18) Lines 246-247: "et al" not "at al"

19) Line 255: "four-fold higher than maternal titer"

20) Line 282: Clarify what sensitivity and specificity you're referencing here. Is it sensitivity and specificity of only the POCT in question? Or were sensitivities and specificities of all comparators taken into consideration, too?

21) Line 315-317: Did Bristow really posit an increase in adverse outcomes? Or it it more accurate to say that Bristow posited no difference in adverse outcomes? Your next paragraph makes it sound like the Bristow reference (#39) really showed no difference. If that's the case, I would tone down the language you use in lines 315-317.

22) Lines 318: Please try to use a consistent number of decimal places as you're reporting findings in this paragraph. It makes sense that for cases you would have 0 decimal places, but for rates per 1,000 pregnancies, try to be consistent.

23) Lines 381-402: The framing of this Discussion section is a little hard to follow. While reading the Results section, I clearly understood that the Bristow paper produced different results than all other modeling studies. I am not certain why the authors are leading with a discussion of the Kruznik papers before discussing Bristow, which seems like the real outlier? It might make the MOST sense to structure your Discussion by walking through model assumptions/inputs (one paragraph on sensitivity/specificity, one paragraph on treatment coverage, one paragraph on syphilis prevalence) and discussing how each paper made different choices leading to different outcomes.

24) Line 464: "...authors' prediction model..." It would also be helpful to refer to this as a treponemal-only POCT vs. a dual POCT in your comparison. ICS feels like an inappropriate term to use, because most dual POCTs are also ICS, right?

25) Line 486: "...poor improvements in testing..." might be better stated as "...modest improvements in testing..." to improve understanding. A poor improvement is a tough oxymoron to interpret.

26) Line 518: Is HIV testing coverage 40-50% higher or 40-50 percentage points higher? Based on my prior work, I suspect it's the latter, so check your word choice here to make sure you're saying what you mean.

27) Line 535: "clinical studies"

28) Line 536: Instead of "implemented in the model" consider: "...used to construct the model."

29) Line 540: Clarify the distinction that you're trying to draw (i.e., some country- and region-level prevalence estimates were generated using treponemal-only methods, which would not indicate a measure of active syphilis infection, but rather a measure of past or current infection.)

30) Line 547-548: I like the fact that you specify reasons why a 100% treatment rate is not a reasonable assumption; I would also add stockouts to the list of reasons. The supply chain for syphilis tests and treatments is not nearly as strong as the supply chain for HIV tests and treatments and stockouts are common, particularly due to lapses in funding for syphilis products. (HIV products rarely experience these lapses in funding).

31) In your limitations section, I would like to see you add/explore the idea that sensitivity and specificity inputs for these modeling studies often use manufacturer-provided sensitivities/specificities, which are indicative of a lab atmosphere in the Western Hemisphere. Often times, 'real world' studies measuring sensitivity and specificity of rapid diagnostic tests used in the field (i.e., Bioline) show significantly lower test characteristics when used by REAL end-users than they do in a lab in Europe (i.e., the manufacturer's insert.) It does make me wonder what TRUE sensitivity/specificity of many of these POCTs would be in the field and how this may impact pregnancy outcome calculations.

32) Line 572: "show" not "shown"

6. PLOS authors have the option to publish the peer review history of their article (what does this mean?). If published, this will include your full peer review and any attached files.

Reviewer #1: No

Reviewer #2: No

Reviewer #3: No

---

## [Author Response · Author response to Decision Letter 0]

28 Jan 2021

We hope we have adequately addressed and responded to all comments. Thank you to the reviewers for their support in improving the quality of our work.

---

## [Editor Report · Decision Letter 1]

11 Feb 2021

The impact of antenatal syphilis point of care testing on pregnancy outcomes: a systematic review

PONE-D-20-29965R1

Dear Dr. Ambrosino,

We’re pleased to inform you that your manuscript has been judged scientifically suitable for publication and will be formally accepted for publication once it meets all outstanding technical requirements.

Kind regards,

Jodie Dionne-Odom, MD

Academic Editor

PLOS ONE
---

## [Editor Report · Acceptance letter]

12 Mar 2021

PONE-D-20-29965R1 

The impact of antenatal syphilis point of care testing on pregnancy outcomes: a systematic review 

Dear Dr. Ambrosino:

I'm pleased to inform you that your manuscript has been deemed suitable for publication in PLOS ONE. Congratulations! Your manuscript is now with our production department. 

Kind regards, 

on behalf of

Dr. Jodie Dionne-Odom 

Academic Editor

PLOS ONE